# Prognostic value of ki67 in BCG-treated non-muscle invasive bladder cancer: a meta-analysis and systematic review

Yuhui He,[1,2] Ning Wang,[3] Xiaofeng Zhou,[1,2] Jianfeng Wang,[2] Zhenshan Ding,[2] Xing Chen,[2] Yisen Deng[1,2]

## ABSTRACT

**Objectives** The aim of this study was to explore the prognostic value of ki67 as a marker in patients with non-muscle invasive bladder cancer (NMIBC) treated with BCG.

**Methods** Studies were systematically retrieved from the relevant databases (Web of Science, PubMed, Cochrane Library and Embase), and the expiry date was May 2017. The research steps referred to the Preferred Reporting Items for Systematic Reviews and Meta-Analysis statement.

**Results** A total of 11 studies that complied with the inclusion criteria were included. The expression of ki67 was not statistically significantly associated with recurrence-free survival (RFS) (HR 1.331; 95% CI 0.980 to 1.809). No significant heterogeneity was found among all included studies ($I^2$=36.7%, p=0.148). The expression of ki67 was statistically significantly associated with progression-free survival (PFS) (HR 2.567; 95% CI 1.562 to 4.219), and the overexpression of ki67 was the risk factor for PFS. Significant heterogeneity was noted among all the included studies ($I^2$=55.6%, p=0.021). The studies that might cause heterogeneity were excluded using the Galbraith plot, and then the meta-analysis was performed again. The results showed that the expression of ki67 was still associated with PFS (HR 2.922; 95% CI 2.002 to 4.266).

**Conclusions** The overexpression of ki67 was the risk factor for PFS, and the relationship between the expression of ki67 and RFS was not statistically significant in patients with NMIBC treated with BCG intravesical immunotherapy. Well-designed, prospective, with a large sample size are still needed to validate the findings.

[1]Peking University China-Japan Friendship School of Clinical Medicine, Beijing, China
[2]Department of Urology, China-Japan Friendship Hospital, Beijing, China
[3]College of Psychology, North China University of Science and Technology, Tangshan, Hebei, China

**Correspondence to**
Dr Xiaofeng Zhou;
doctorzxf@126.com

## INTRODUCTION

Bladder cancer is one of the most common clinical urological tumours. It is a direct threat to the survival of patients with the disease. The incidence of bladder cancer varies across the world, with the highest rate in the developed communities.[1] There was a total of 429 800 new cases of bladder cancer and 165 100 deaths worldwide in 2012. Bladder cancer occurs mostly in men, and about 10-fold variation in incidence rates has been reported internationally.[2] About 70% of these patients have non-muscle

invasive bladder cancer (NMIBC).[3] Four major organisational guidelines on NMIBC, including the American Urological Association/Society of Urologic Oncology, the European Association of Urology, the National Comprehensive Cancer Network, and the National Institute for Health and Care Excellence guidelines, recommend that proper initial transurethral resection (TUR) of bladder tumour is a critical step in the initial management and staging of the disease.[4] However, TUR surgery alone cannot solve the postoperative problems of NMIBC because of high recurrence rate and disease development.[5] Postoperative TUR associated with BCG intravesical immunotherapy can prevent postoperative recurrence of NMIBC and significantly reduce the moderate and high risk of development of NMIBC.[6 7] However, postoperative BCG intravesical immunotherapy still has some problems. The failure rate of BCG intravesical therapy in NMIBC is about 40%–50%.[8] Furthermore, BCG has toxic side effects, such as hepatitis, pneumonitis, epididymitis/orchitis, abscess formation,

**Strengths and limitations of this study**

► This meta-analysis and systematic review was performed via a strict literature search. It was the first meta-analysis to evaluate the prognostic value of ki67 in patients with NMIBC after transurethral resection and BCG intravesical immunotherapy.
► The number of studies considered in the final meta-analysis was 11. This small sample size limited the potential analyses. The research did not consider the surgical skills mentioned in published studies.
► Despite a systematic search strategy, the inclusion criteria excluded non-English documents and had language bias. The meta-regression analysis suggested no bias, but a selection bias was likely.
► These limitations notwithstanding, the research can guide the follow-up research on immunohistochemical markers and clinical practice in NMIBC.

bladder contracture, ureteral obstruction, BCG sepsis, leucopenia and haematuria.[9] Therefore, BCG therapy should be individually performed, and the patients in whom BCG therapy is ineffective should be recognised in time. These patients or those with poor prognosis should receive radical cystectomy or any other therapy in time to avoid futile treatment and to alleviate pain. However, it is still difficult to recognise patients with no effect of TUR postoperative BCG intravesical immunotherapy due to the heterogeneity of bladder cancer and the individuality of patients.[10] Therefore, it is necessary to find the prognostic factors for patients with NMIBC undergoing TUR and receiving BCG therapy.

The recurrence rate of bladder cancer treated with different therapies is between 50% and 80%, and about 15% of the low-grade tumour recurrence involves high-grade tumours.[11] Cystoscopy should be performed periodically in patients so that the recurrent lesion can be detected in time. A reliable prognostic molecular marker can reduce the pain caused by cystoscopy. The absence of reliable prognostic markers for NMIBC, makes it hard to decide on the postoperative therapy in the clinic,[12] which depends mainly on clinical guidelines and the physician's experience. Currently, some of the published studies on immunohistochemical markers have evaluated the prognostic value of BCG intravesical immunotherapy on the patients receiving TUR first. The main immunohistochemical markers include ki67, p53, p27, pRb, CD9, CD20, E2F1 and so on.[13 14] However, no immunohistochemical marker has been confirmed so far. The prognostic value of the ki67 antigen for the survival of patients with NMIBC receiving BCG intravesical immunotherapy has been controversial. For example, Kruger[15] reported that the ki67 antigen was an independent predictive factor for the recurrence of pT1 stage tumour, but Oderde[16] believed that ki67 was an independent predictive factor for the recurrence of all NMIBCs. Zlotta[17] reported that ki67 antigen had no independent prognostic value in patients receiving BCG therapy. Saint[18] retrospected on the recent 25-year published studies and believed that the independent prognostic factor for bladder cancer in patients receiving BCG therapy was not clear. An international consensus group listed various bladder cancer prognostic indexes by reviewing PubMed and considered that although some markers (such as ki67 and p53) could predict the recurrence and development of bladder cancer, the data still had heterogeneity. Thus, strict test criteria and clear statistical methods should be established for further evaluation.[19]

A meta-analysis can enlarge the sample size by integrating independent studies with small sample size, further increase the statistical efficacy and reduce the wrong conclusion caused by the small sample size.[20] The aim of this study was to explore the prognostic value of ki67 as a marker in patients with NMIBC treated with BCG. Based on the literature search, this study was the first meta-analysis to evaluate the prognostic value of ki67 in patients with NMIBC treated with BCG.

## METHODS

This meta-analysis was performed according to the Preferred Reporting Items for Systematic Reviews and Meta-Analyses (PRISMA) statement (online supplementary table S1).[21] The present meta-analysis did not need approval because all the included published studies were approved by the ethics committees in the respective research institutes.

### Literature retrieval strategy

The comprehensive literature search was performed on Web of Science, PubMed, Cochrane Library and Embase databases for relevant studies. The last quest was updated on 24 May 2017, with hand-searching to identify any potentially eligible studies that might have been missed. The following search strategy was adopted for each database: ("Urinary Bladder Neoplasms"(Mesh) OR "bladder cancer" OR "bladder carcinoma" OR "bladder tumour") AND ("BCG Vaccine"(Mesh) OR "BCG" OR "Bacillus Calmette–Guérin") AND ("ki67 antigen"(Mesh) OR "ki-67" OR "ki67" OR "MBI-1"). Filters were as follows: retrospective, array research, clinical trial, controlled clinical trial and randomised controlled trial. Free word retrieval strategy was used. The contents included the reference lists and relevant suggestive references while searching (online supplementary file S1).

### Inclusion and exclusion criteria

The inclusion criteria were as follows: (1) Prospective or retrospective published studies evaluating the prognostic relationship between the expression of ki67 and NMIBC treated with BCG. (2) The expression of ki67 in tissues detected by immunohistochemistry analysis. (3) HRs and 95% CIs directly obtained from the published studies. (4) Published English studies. The exclusion criteria were as follows: (1) Review, systematic evaluation, case report, editorial and specialist experience. (2) Studies with no human subjects. (3) Published studies in which data could not be extracted or those having wrong data.

### Data extraction and evaluation of literature quality

Based on the aforementioned criteria, two reviewers independently screened the published studies by reading titles and abstracts and got preliminary conclusions. If the conclusions were not consistent, the literature was discussed by all the authors to decide on its inclusion. Relevant information from the included published studies, such as first author, publication time, research country, sex, case number, age, follow-up date, disease stage, cut-off values, recurrence-free survival (RFS) and progression-free survival (PFS), was extracted. The Newcastle-Ottawa

Scale (NOS) was used to evaluate the quality of all the published studies.[22] Scores 0–3, 4–5 and 6–8 were accepted as low quality, medium quality and high quality, respectively.

## Statistical methods

The measuring time and method of ki67 complied with the standard of clinical routine and pathological examination. Tumour tissue samples were taken in accordance with the standard surgical procedure and used for immunohistochemical analysis. RFS and PFS were the traditionally used statistical parameters. PFS was defined as the time from the beginning of treatment to the first progression. RFS was defined as the time from the removal of the lesion (or the randomisation of the clinical trial) until the recurrence or death of the tumour. The impact of the expression of ki67 on survival was quantified using the combined HRs and 95% CIs. The HR and 95% CI of each study were directly extracted from the original published study. Besides, Parmar and Tierney's[23] method was used to extract the data because some of the published studies did not directly provide HR and 95% CI. For example, some studies provided only the survival curve. In this meta-analysis, the DerSimonian-Laird random-effects model[24] was used because only the random-effects model is suitable for large heterogeneity. Similar to traditional methods, HR>1 was considered as the prognostic risk factor for the overexpression of ki67, and HR<1 was a protective factor. A 95% CI<1 indicated a statistically significant difference in the relationship between the overexpression of ki67 and prognosis.

Heterogeneity was calculated according to $\chi^2$-based Q test and $I^2$ statistic,[25] and was judged using the $I^2$ value (low heterogeneity: $I^2$<25%; moderate heterogeneity: $I^2$=25%–50%; large heterogeneity: $I^2$>50%). Besides, a p value >0.05 was considered as low heterogeneity. Then, subgroup analysis based on regions, sample size, follow-up period, tumour grading, cut-off value, publication time and patient age was performed. A value of 1% was considered to be statistically significant in the subgroup analysis. A Galbraith plot was used to search published studies with heterogeneity,[26] and the meta-analysis was again performed after excluding these published studies. Meanwhile, the factors causing heterogeneity were also explored using the residual maximum likelihood based random-effects meta-regression analysis.[27] All the statistical analyses were performed using Stata V.12.0 software (StataCorp, College Station, Texas, USA), and two-sided test was used to evaluate the p value.

## Estimation of publication bias

Begg's plot and Egger's test method were used to find possible publication bias. A p value <0.05 was considered to indicate publication bias.

## RESULTS

### Literature screening

A total of 97 published studies were retrieved. Furthermore, 68 of them were excluded after duplicates were removed and records screened, and 18 were excluded after reading the full text (10 published studies from which HR and 95% CI could not be obtained, 2 non-English studies and 6 that did not use ki67 detection). Finally, 11 published studies were included in the meta-analysis (figure 1).

### Basic characteristics and quality evaluation of included published studies

The 11 included studies were published between 1997 and 2013, and the countries included Italy, South Korea, Spain, Germany, New Zealand, Canada, Portugal and France. The largest sample size was 309, and the smallest was 32. A total of 1321 patients were enrolled in this study. The follow-up period was more than 36 months, and the longest was 229 months. T1 was the main tumour grading, and the cut-off value ranged from 10.4% to 40%. Seven published studies reported patients' RFS, and nine reported PFS (table 1). One literature was scored as six stars by NOS, seven as seven stars and three as eight stars. The median of the NOS Score was 7 (table 2).

### Influence of the expression of ki67 on RFS

Seven published studies reported the expression of ki67 and PFS results of patients with NMIBC treated with BCG. The meta-analysis indicated that ki67 had no statistically significant association with RFS (HR 1.331; 95% CI 0.980 to 1.809), and no heterogeneity among the included studies was reported ($I^2$=36.7%, p=0.148) (figure 2A). The subgroup analysis was performed based on the regions, sample size, follow-up period, stage, cut-off value, publication time and age. Meanwhile, all the original published analyses on the association between the expression of ki67 and RFS in patients with NMIBC treated with BCG were multivariate, and the HRs were adjusted. The stratification analysis by region indicated that ki67 was also significantly associated with RFS in Caucasians and a follow-up period shorter than 60 months (HR 1.441, 95% CI 1.014 to 2.047; HR 1.853, 95% CI 1.316 to 2.607) (table 3).

### Influence of the expression of ki67 on PFS

A total of nine published studies reported the expression of ki67 and PFS results of patients with NMIBC treated with BCG. The meta-analysis indicated that ki67 had no statistically significant association with RFS (HR 2.567, 95% CI 1.562 to 4.219), and the overexpression of ki67 was the risk factor for PFS. Statistically significant heterogeneity was found among all the included studies ($I^2$=55.6%, p=0.021) (figure 2B). The subgroup analysis was performed based on the regions, sample size, follow-up period, stage, cut-off value, publication time and age. However, the data extracted from six original

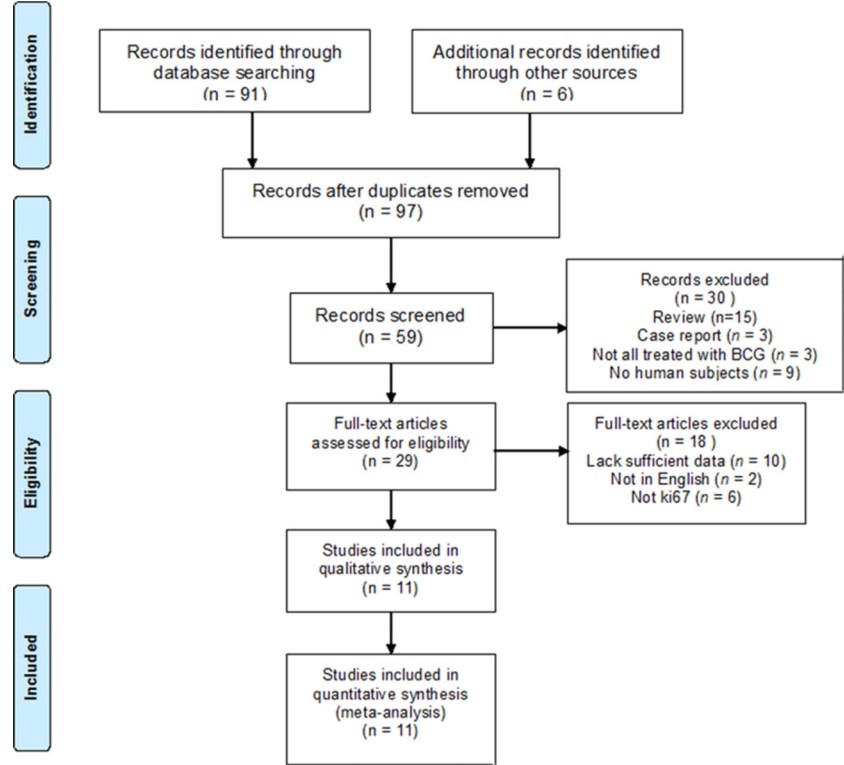

**Figure 1** Flow diagram of study selection.

published analyses on the association between the expression of ki67 and PFS in patients with NMIBC treated with BCG were multivariate with adjusted HRs, whereas the data from three original published analyses were univariate with unadjusted HRs. In the stratified analyses by region, sample size, follow-up time, stage, cut-off, publication year and patient age, significant associations were observed in the studies with the Caucasian subgroup, sample size >100, follow-up period <60 months, other cut-offs and two subgroups based on age (HR 1.97, 95% CI 1.04 to 3.74; HR 2.37, 95% CI 1.23 to 4.55; HR 2.49, 95% CI 1.19 to 5.21; HR 2.515, 95% CI 1.382 to 4.576; HR 2.800, 95% CI 1.447 to 5.418; and HR 2.654, 95% CI 1.381 to 5.100, respectively). However, significant associations were also observed in both multivariate and univariate analyses (HR 2.10, 95% CI 1.07 to 1.12; HR 2.80, 95% CI 1.65 to 7.85, respectively), and the effect size suggested the same outcomes (HR 2.567, 95% CI 1.562 to 4.219) (table 4).

### Galbraith plot
Using Galbraith plot (figure 3A), it was found that the study by Santos[28] was the main reason for the heterogeneity of RFS. After the aforementioned study was

| Table 1 | | Main characteristics of all studies included in this meta-analysis | | | | | | | |
|---|---|---|---|---|---|---|---|---|---|
| Study | Year | Country | Male/ female | No. of patients | Age (years) | Follow-up (month) | Stage | Cut-off | Survival analysis |
| Oderda[16] | 2013 | Italy | 166/26 | 192 | 73.2 (SD 11.9) | 100 (2–229) | All NMIBC | 20% | RFS |
| Park[14] | 2013 | Korea | 53/8 | 61 | 66 (31–85) | 60 (6–217) | T1G3 | 10.4% | RFS/PFS |
| Quintero[45] | 2013 | Spain | 143/21 | 164 | 61 (29–93) | 75 (60–144) | Ta | 13% | PFS |
| Bertz[12] | 2012 | Germany | 237/72 | 309 | 71.7 (38–87) | 49 (5–172) | pT1 | 15% | RFS/PFS |
| van Rhijn[29] | 2012 | The Netherlands, Canada | 105/24 | 129 | 68.8 (SD 9.9) | 78 (39.6–110.4) | T1 | 25% | RFS/PFS |
| Burger[46] | 2007 | Germany | 45/21 | 71 | 71 (52–94) | 39 (1–133) | T1/Ta | 15% | RFS |
| Queipo-Zaragoza[47] | 2007 | Spain | 71/12 | 83 | 68.1 (SD 8.5) | All>36 | T1G3 | 40% | PFS |
| Lopez-Beltran[48] | 2004 | Spain | 49/2 | 51 | 69.96 (49–89) | 63.82 (60–144) | T1G3 | 13% | PFS |
| Santos[28] | 2003 | Portugal | 115/44 | 159 | 66 (21–88) | 46.5 (4–123) | pTa/pT1 | 18% | RFS/PFS |
| Blanchet[49] | 2001 | France | - | 70 | 62.6 (21–84) | 64 (12–111) | pT1/pTa | 13% | PFS |
| Lee[50] | 1997 | Korea | 28/4 | 32 | 57.1 (30–81) | All >24 | T1G2–3 | 20% | RFS |

NMIBC, non-muscle invasive bladder cancer; no., number; PFS, progression-free survival; RFS, recurrence-free survival.

**Table 2** Quality of the included studies assessed by the Newcastle-Ottawa Scale

| Study | Selection | | | | Comparability | Exposure | | | Scores |
| | Adequate definition of cases | Representativeness of cases | Selection of controls | Definition of controls | Control for important factor | Ascertainment of exposure | Same method to ascertain as for cases and controls | Non-response rate | |
| --- | --- | --- | --- | --- | --- | --- | --- | --- | --- |
| Oderda[16] | – | ☆ | ☆ | ☆ | ☆☆ | ☆ | ☆ | ☆ | 8 |
| Park[14] | – | ☆ | ☆ | – | ☆☆ | ☆ | ☆ | ☆ | 7 |
| Quintero[45] | – | ☆ | ☆ | – | ☆☆ | ☆ | ☆ | – | 6 |
| Bertz[12] | – | ☆ | ☆ | – | ☆☆ | ☆ | ☆ | ☆ | 7 |
| van Rhijn[29] | – | ☆ | ☆ | – | ☆☆ | ☆ | ☆ | ☆ | 7 |
| Burger[46] | ☆ | ☆ | ☆ | – | ☆☆ | ☆ | ☆ | – | 7 |
| Queipo-Zaragoza[47] | – | ☆ | ☆ | – | ☆☆ | ☆ | ☆ | ☆ | 7 |
| Lopez-Beltran[48] | – | ☆ | ☆ | ☆ | ☆☆ | ☆ | ☆ | ☆ | 8 |
| Santos[28] | – | ☆ | ☆ | – | ☆☆ | ☆ | ☆ | ☆ | 7 |
| Blanchet[49] | – | ☆ | ☆ | ☆ | ☆☆ | ☆ | ☆ | ☆ | 8 |
| Lee[50] | – | ☆ | ☆ | – | ☆☆ | ☆ | ☆ | ☆ | 7 |

excluded, the remaining RFS studies had no significant heterogeneity according to the new meta-analysis ($I^2$=0.0%, p=0.667). However, the expression of ki67 still had no statistically significant association with RFS (HR 1.161, 95% CI 0.896 to 1.504) (online supplementary figure S1). Using the Galbraith plot (figure 3B), it was found that the Santos,[28] Park[14] and van Rhijn[29] studies were the main reason for the heterogeneity of PFS. After these studies were excluded, the remaining RFS studies had no significant heterogeneity according to the new meta-analysis ($I^2$=0.0%, p=0.497). The expression of ki67 still had a statistically significant association with PFS (HR 2.922, 95% CI 2.002 to 4.266) (online supplementary figure S2).

### Meta-regression analysis
The meta-regression analysis indicated that the factors influencing heterogeneity (publication time, research region, sample size, stage, cut-off value, age and follow-up period) might not be the reason for RFS heterogeneity. Publication time was the reason for heterogeneity of PFS (P=0.036), but other factors were not (online supplementary table S2).

### Publication bias
The funnel plot (figure 4A) was basically symmetrical for RFS. The results of Begg's test and Egger's test showed p=0.761 (Begg's) and p=0.601 (Egger's). The funnel plot (figure 4B) was also basically symmetrical for PFS. The results of Begg's test and Egger's test showed p=0.917 (Begg's) and p=0.964 (Egger's).

### DISCUSSION
A total of 11 published studies with 1321 cases complying with the inclusion criteria were included in this meta-analysis. The results of the meta-analysis indicated that the expression of ki67 had no statistically significant association with RFS, but it was significantly associated with PFS. The overexpression of ki67 was the risk factor for PFS. It suggested that ki67 was the prognostic predictive marker in patients with NMIBC treated with BCG. Besides, the aforementioned conditions did not change after excluding the published studies, possibly leading to heterogeneity and reperforming of the meta-analysis. It further proved that the result of the aforementioned meta-analysis was stable, that is, the overexpression of ki67 was the risk factor for PFS. In the Caucasian subgroup for PFS, racial classification and regional factors might be crucial in the prognosis of patients with NMIBC after BCG therapy. This might be related to the existence of different drug gene susceptibilities in people belonging to different races and living areas. The two subgroups were based on age in PFS, suggesting that age might be the important factor influencing the prognosis of bladder cancer. This also complies with our clinical practice. The elder the patient, the worse the prognosis. Several factors led to heterogeneity in the aforementioned subgroup analysis: (1) Due to the influence of race and environment, the documents included in this study came from different regions and countries. A large number of studies confirmed the differences in disease susceptibility between ethnic groups and regions. (2) Differences existed in the operation of healthcare workers in TUR and BGC intravesical immunotherapy because of different regions and different clinicians, such as surgical clearance of the tumour. It is not often easy to completely remove a tumour with a broad-based surface and also depends on the surgeon's experience and surgical skills. In addition, the quality of BCG manufacturers may vary from region to region. (3) Different literature might include the bias of research object, research design, measuring instrument and so on. However, in general,

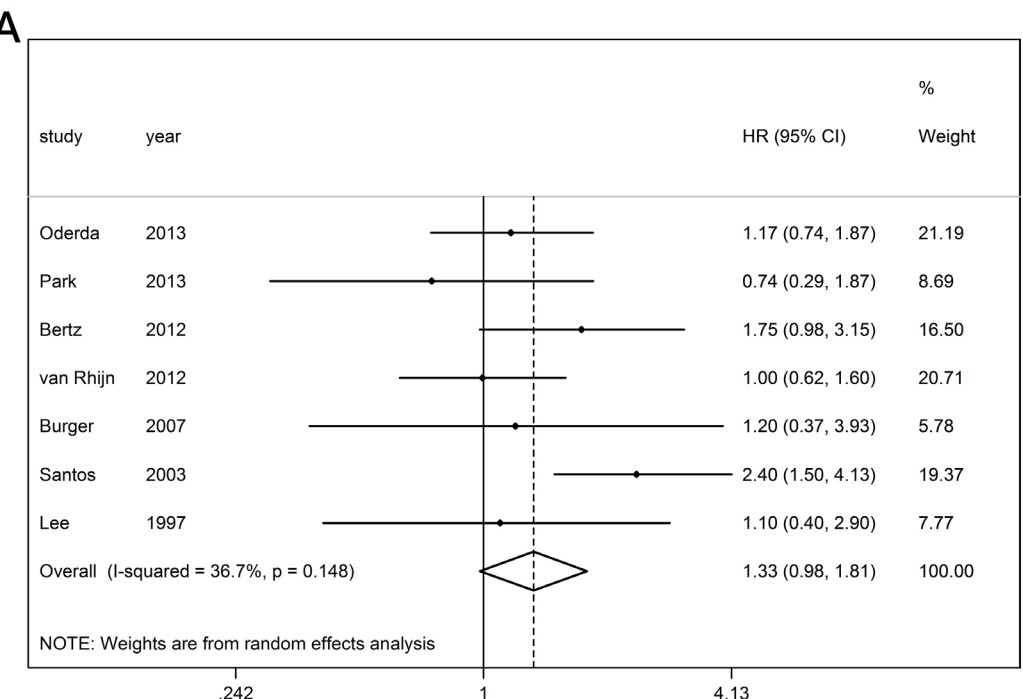

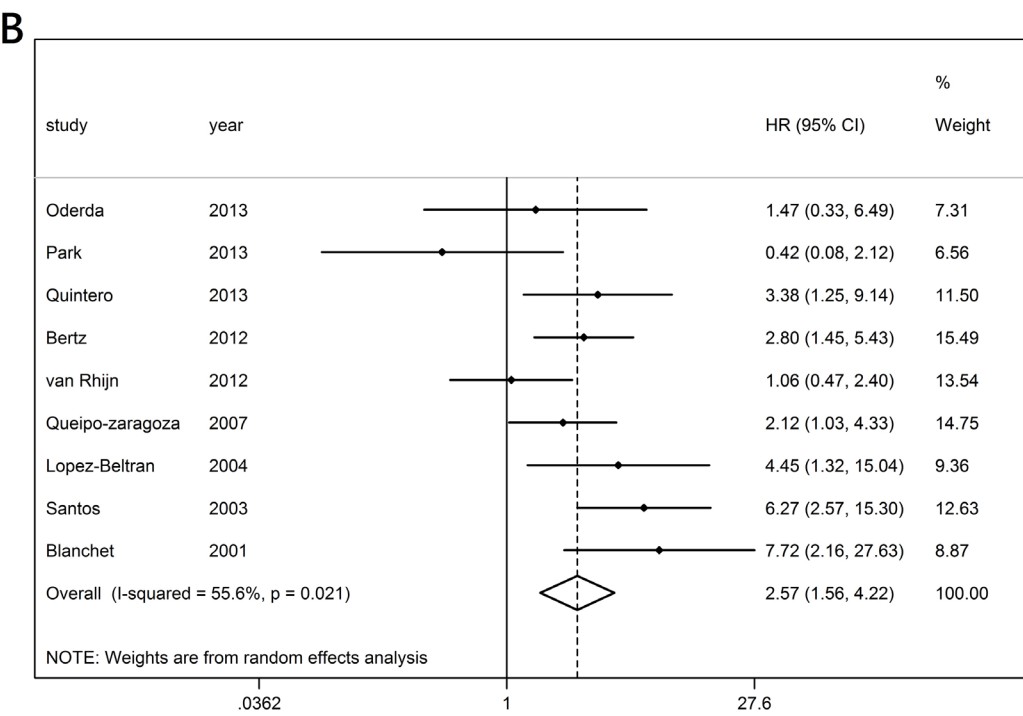

**Figure 2** Forest plots of HRs estimated for the relationship between the expression of ki67 and recurrence-free survival (A) or progression-free survival (B) among patients with non-muscle invasive bladder cancer treated with BCG.

heterogeneity did not affect the conclusion. Besides, the meta-regression analysis indicated publication time as the reason for PFS heterogeneity. It might be related to the improvement in testing technology, research level, and the quality and number of published studies, facilitating follow-up studies. As all the original data extracted from published studies on the association between the expression of ki67 and RFS in patients with NMIBC treated with

BCG were multivariate, the result was considered to be precise because the HRs were adjusted, excluding the confounding factors such as age and gender. However, the original data extracted from published analyses on the association between the expression of ki67 and PFS were both multivariate and univariate. It was believed that the aforementioned adjustments did not have a significant impact on meta-analyses. Besides, according to the

**Table 3** Subgroup results of RFS and heterogeneity test

| Variables | Analysis number | HR (95% CI) | Heterogeneity test | | |
|---|---|---|---|---|---|
| | | | Q | P | $I^2$ (%) |
| Total RFS | 7 | 1.331 (0.980 to 1.809) | 9.48 | 0.148 | 36.7 |
| Region | | | | | |
| Asian | 2 | 0.892 (0.454 to 1.752) | 0.32 | 0.570 | 0.0 |
| Caucasian | 5 | 1.441 (1.014 to 2.047) | 7.52 | 0.111 | 46.8 |
| Sample size | | | | | |
| >100 | 4 | 1.466 (0.986 to 2.181) | 7.44 | 0.059 | 59.7 |
| ≤100 | 3 | 0.959 (0.534 to 1.725) | 0.51 | 0.777 | 0.0 |
| Follow-up (month) | | | | | |
| ≥60 | 3 | 1.036 (0.758 to 1.415) | 0.79 | 0.674 | 0.0 |
| <60 | 4 | 1.853 (1.316 to 2.607) | 2.62 | 0.453 | 0.0 |
| Stage | | | | | |
| All NMIBC | 3 | 1.575 (0.915 to 2.711) | 4.44 | 0.109 | 54.9 |
| Others | 4 | 1.153 (0.821 to 1.620) | 3.21 | 0.360 | 6.6 |
| Cut-off | | | | | |
| 15% | 2 | 1.625 (0.963 to 2.743) | 0.31 | 0.575 | 0.0 |
| Others | 5 | 1.252 (0.839 to 1.869) | 8.56 | 0.073 | 53.3 |
| Publication year | | | | | |
| ≥2012 | 4 | 1.164 (0.874 to 1.550) | 3.20 | 0.362 | 6.3 |
| <2012 | 3 | 1.774 (1.046 to 3.008) | 2.57 | 0.277 | 22.1 |
| Patient age (year) | | | | | |
| ≥70 | 3 | 1.352 (0.955 to 1.913) | 1.16 | 0.559 | 0.0 |
| <70 | 4 | 1.256 (0.717 to 2.198) | 8.32 | 0.040 | 63.9 |

NMIBC, non-muscle invasive bladder cancer; RFS, recurrence-free survival.

funnel plot, Begg's test and Egger's test, the included studies had no statistically significant publication bias. Thus, the reliability of the present meta-analysis was high.

In 2016, the European Association of Urology[30] recommended a scoring system for the prognostic evaluation of NMIBC based on six clinical and pathological factors proposed by the European Organisation for the Research and Treatment of Cancer (EORTC) Genito-Urinary Group, including number of tumours, tumour size, prior recurrence rate, T category, presence of concurrent carcinoma in situ and tumour grade (online supplementary table S3). The tumours were categorised into low-risk, intermediate-risk and high-risk using this assessment system to evaluate the prognosis. For the patients after BCG therapy, the European Association of Urology recommended another risk calculator developed by Club Urologico Espanol de Tratamiento Oncologico (CUETO) and EORTC. This calculator was based on gender, age, recurrent tumour, number of tumours, T category, associated Tis and grade. The CUETO risk calculator can be found at http://www.aeu.es/Cueto.html. The recommended level was B grade for the two scales for patients with NMIBC, whether used alone or combined. The two scales could be used together in the clinic. When using the CUETO Scale, the calculated recurrent risk was lower than that from the EORTC Scale,[31] which might be related to the special design in the CUETO Scale for the patients receiving BCG intravesical immunotherapy. However, the scoring system that depended only on clinical and pathological factors could not accurately evaluate the prognosis of patients with bladder cancer in the T1 stage due to the independence of disease condition in each patient.[32] The markers regulated at the genetic level may judge the prognosis of patients with bladder cancer with the development of precision medicine. A reliable marker helps recognise, in time, the high-risk patients who have failed BCG intravesical immunotherapy. Hence, these patients can undergo radical cystectomy or other treatments, in time. Unfortunately, no prognostic marker has currently been applied in the clinic. The results of this study potentially help to remind clinicians that patients with high expression of ki67 may need to develop more personalised follow-up plans, such as shorter follow-up and cystoscopy cycles. Patients with high risk of clinical evaluation of the guidelines and overexpression of ki67 may need to promptly change the treatment strategy.

Ki67 is a nucleoprotein that can be detected in the cell cycles except during the G0 phase.[33] The expression of human ki67 protein is closely related to proliferation. Therefore, it is an ideal marker to confirm the growth

**Table 4**  Subgroup results of PFS and heterogeneity test

| Variables | Analysis number | HR (95% CI) | Heterogeneity test | | |
| --- | --- | --- | --- | --- | --- |
| | | | Q | P value | $I^2$ (%) |
| Total PFS | 9 | 2.567 (1.562 to 4.219) | 18.10 | 0.021 | 55.6 |
| Region | | | | | |
| Asian | 1 | 0.421 (0.084 to 2.114) | 0.00 | | |
| Caucasian | 8 | 2.883 (1.830 to 4.544) | 12.99 | 0.072 | 46.1 |
| Sample size | | | | | |
| >100 | 5 | 2.559 (1.372 to 4.774) | 9.26 | 0.055 | 56.8 |
| ≤100 | 4 | 2.536 (0.943 to 6.818) | 8.75 | 0.033 | 65.7 |
| Follow-up (month) | | | | | |
| ≥60 | 6 | 2.153 (0.984 to 4.710) | 13.08 | 0.023 | 61.8 |
| <60 | 3 | 3.158 (1.774 to 5.623) | 3.56 | 0.169 | 43.8 |
| Stage | | | | | |
| All NMIBC | 3 | 4.673 (1.938 to 11.264) | 3.29 | 0.193 | 39.2 |
| Others | 6 | 2.044 (1.213 to 15.040) | 9.58 | 0.088 | 47.8 |
| Cut-off | | | | | |
| 15% | 1 | 2.800 (1.447 to 5.418) | 0.00 | | |
| Others | 8 | 2.515 (1.382 to 4.576) | 17.92 | 0.012 | 60.9 |
| Publication year | | | | | |
| ≥2012 | 5 | 1.685 (0.883 to 3.215) | 8.04 | 0.090 | 50.2 |
| <2012 | 4 | 4.176 (2.209 to 7.884) | 5.00 | 0.172 | 40.0 |
| Patient age (year) | | | | | |
| ≥70 | 2 | 2.519 (1.377 to 4.606) | 0.60 | 0.438 | 0.0 |
| <70 | 7 | 2.654 (1.381 to 5.100) | 17.40 | 0.008 | 65.5 |
| Multivariate/univariate | | | | | |
| Multivariate | 6 | 2.101 (1.070 to 1.121) | 13.83 | 0.031 | 63.8 |
| Univariate | 3 | 2.803 (1.652 to 7.856) | 3.38 | 0.001 | 40.8 |

NMIBC, non-muscle invasive bladder cancer; PFS, progression-free survival.

fraction of specific cell colonies.[34] Ki67 is a widely known amplified biomarker. The ki67 monoclonal antibody can be detected by the immunohistochemical method.[35] Ki67 has been proved to be a good proliferation marker in different cancers, including bladder cancer.[36]

So far, some meta-analyses have studied the effect of ki67 on the prognostic quality of life of patients with oesophageal cancer, breast cancer, epithelial ovarian cancer and so on.[37–39] Some studies have also focused on other aspects of bladder cancer. Using meta-analysis, Luo[40] believed that a high reactivity of ki67 could predict the poor prognosis in patients with bladder cancer. The univariate analysis showed that cancer-specific survival, disease-free survival, overall survival, PFS and RFS had a significant association with poor prognosis in patients with a high reactivity of ki67. However, this study included all types of bladder tumours and all the therapies for NMIBC. Currently, the bladder cancer treated in the clinic is mainly NMIBC. Thus, most of the applied therapy is TUR combined with installations of chemotherapy or BCG intravesical immunotherapy based on the patients'

conditions. Therefore, this analysis had a certain limitation in the prognosis of patients with NMIBC after BCG intravesical immunotherapy.

Currently, few evidence-based studies focused on the prognosis of patients with NMIBC after BCG intravesical immunotherapy. Using meta-analysis, Zhou[41] analysed the association between the expression of p53 and quality of life of patients with NMIBC after BCG intravesical immunotherapy. They believed that the overexpression of p53 in patients with NMIBC treated with BCG might be associated with RFS, especially in the Asian population. Similarly, Du[42] also carried out a meta-analysis on the relationship between p53 status and NMIBC in the T1 stage and believed that the overexpression of p53 might be related to the development of NMIBC. The present study indicated that the overexpression of ki67 was the risk factor for PFS, but the expression of ki67 had no statistically significant association with RFS. P53 is the most common inactivated tumour suppressor gene in tumour cells.[43] The inactivation of p53 may cause cell abnormal hyperplasia and cancerisation. The variation in p53 results in enhanced proliferation, invasion

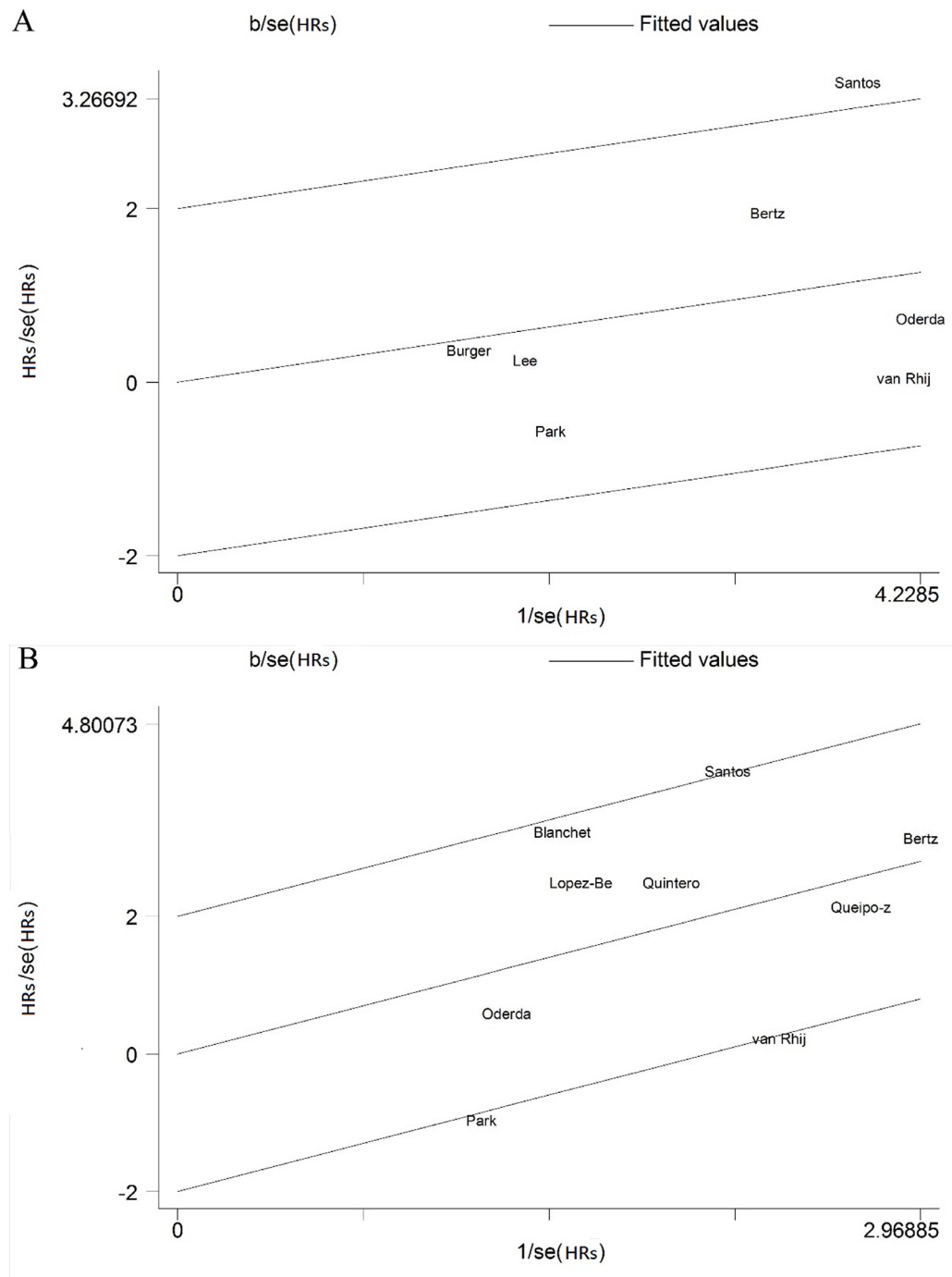

**Figure 3** Galbraith plot analysis was used to evaluate heterogeneity and recurrence-free survival (A) or progression-free survival (B).

and metabolism.[44] The increase in the expression of ki67, as a cell proliferation marker suggests enhanced proliferation.[34] As a tumour suppressor gene with complicated function, p53 has a wider range of effects. The accuracy in the prediction of quality of life may not be more appropriate compared with ki67. The genetic difference between Asians and Caucasians suggests that different prediction systems should be built for different races. Besides, p27, E2F1, ezrin and CK20 were also studied in other investigations for predicting NMIBC prognosis, which could be explored further comparing the advantages of using them alone or combined.

However, this study still had some limitations. First, the published studies included involved different populations, used similar detection equipment and had different cut-off values. All these reasons might have led to heterogeneity. Further, the sample size of the meta-analysis also limited its significance. Second, the meta-analysis included published English studies. Although Begg's test and Egger's test did not suggest publication bias, this study was still influenced by some bias. Finally, the surgical skills were different in the different studies, affecting judgement regarding the effectiveness of BCG.

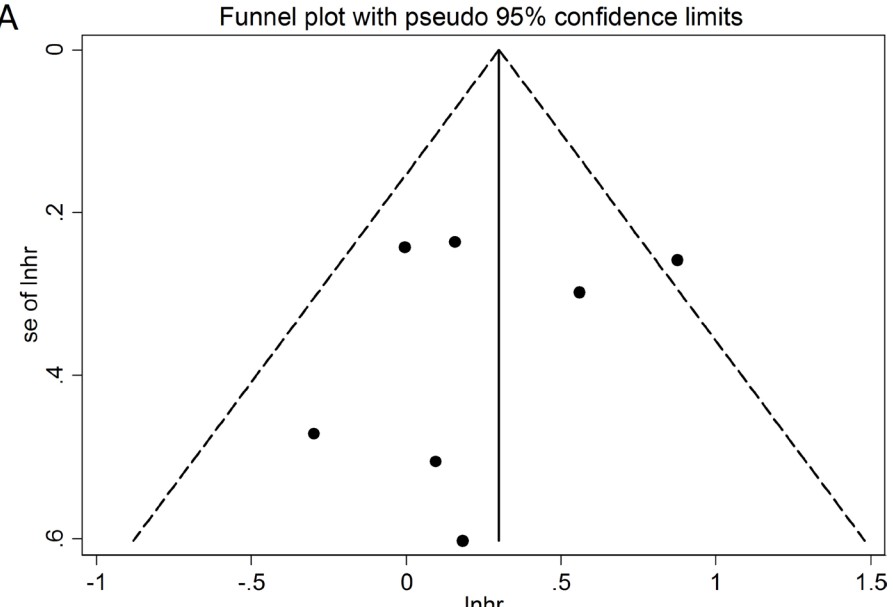

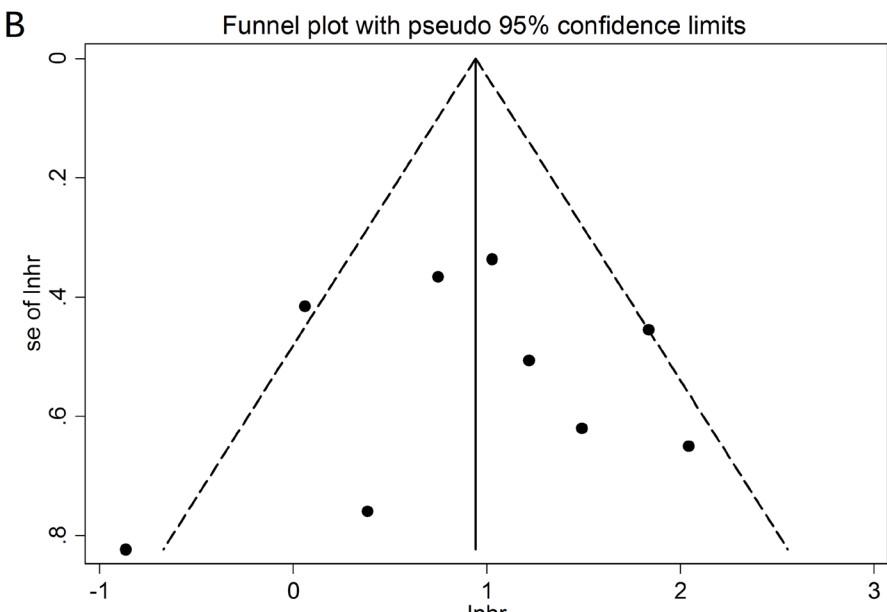

**Figure 4** Funnel plots of the expression of ki67 and recurrence-free survival (A) or progression-free survival (B).

## CONCLUSIONS

The overexpression of ki67 was the risk factor for PFS in patients with NMIBC after TUR and BCG intravesical immunotherapy, but the relationship between the expression of ki67 and RFS was not statistically significant. Due to the aforementioned limitations of the present study, randomised controlled trials with a large sample size are still required to validate the results.

**Contributors** YH, NW and XZ conceived and designed the experiments. XC and YD extracted the data. YH, NW and XZ analysed the data. ZD, JW and XC contributed reagents/materials/analysis tools. YH and NW wrote the paper. XZ critically revised the report.

**Funding** This study was supported by grants from the International S&T Cooperation Program of China (ISTCP) (Grant No.2014DFE30010).

**Disclaimer** The contents of the present study are solely the responsibility of the author. The funders had no role in study design, data collection and analysis, decision to publish, or preparation of the manuscript.

**Competing interests** None declared.

**Patient consent** Not required.

**Provenance and peer review** Not commissioned; externally peer reviewed.

**Data sharing statement** Extra data can be accessed via the Dryad data repository at http://datadryad.org/ with https://doi.org/10.5061/dryad.hf06q72.

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
