## [Reviewer comments · BMJ Open]

ARTICLE DETAILS

TITLE (PROVISIONAL)	Prognostic Value of ki67 in Bacillus Calmette–Guérin-Treated Non-muscle-Invasive Bladder Cancer: a Meta-analysis and Systematic Review
AUTHORS	He, Yuhui; Wang, Ning; Zhou, Xiaofeng; Wang, Jianfeng; Ding, Zhenshan; Chen, Xing; Deng, Yisen

VERSION 1 – REVIEW

REVIEWER	Hanjong, Ahn Asan Medical Center, University of Ulsan College of Medicine, Seoul, Republic of Korea
REVIEW RETURNED	02-Oct-2017

GENERAL COMMENTS	Thank you for an interesting paper about prognostic value of ki67. There's nothing to comment about.
--

REVIEWER	Dr. Ajili Faouzia Laboratory of Human and Experimental Pathology Pasteur Institute of Tunis, Tunisia
REVIEW RETURNED	04-Oct-2017

GENERAL COMMENTS	I have no comment for this paper.
-----------------------------------

REVIEWER	Kentaro Sakamaki The University of Tokyo, Japan
REVIEW RETURNED	28-Oct-2017

GENERAL COMMENTS	General comment; This manuscript described results of a meta-analyses to explore whether ki67 is a prognostic factor of recurrence free survival (RFS) and progression free survival (PFS) in Bacillus Calmette–Guérin-treated non-muscle-invasive bladder cancer. Although the authors concluded that ki67 was not a prognostic factor of RFS but of PFS, several aspects of this manuscript should be improved to conclude that.
---

	Major comments;  1. Hazard ratios (HRs) of univariate analysis or of multivariate analysis were used in the meta-analysis. The authors should clarify which HRs are unadjusted or adjusted in the manuscript, consider an influence of adjustments in meta-analysis and interpret results appropriately. 2. It is unclear how factors in subgroup analyses and meta-regression were selected. For example, 'Cut off' was binarized as '15%' and 'Others', however 'Others' included such as '13%' and '40%'. The authors seemed not to consider 'Publication Time' in subgroup analyses but in meta-regression. And almost prognostic factors were not mentioned in the manuscript. Selection of factors depends on the HRs were adjusted or not. Please explain the validity of the subgroups analysis and the meta-regression. 3. In order that HRs were valid, it is important for when and how ki67 was measured. The definitions of RFS and PFS are also important. If there was any difference, please clarify the measurements and the definitions in Methods and Results sections. 4. Heterogeneity of HRs would be caused by many reasons, including Comment 1-3. The discussion on the heterogeneity would be poor. Please explain whether the heterogeneity could change the conclusion. Minor comments;  1. Axes of Figure 4 were not correct. Please correct these axes. 2. Reference 27 is not appropriate. Please change the reference appropriately.
--	--

REVIEWER	Jonathan Gelfond UT Health San Antonio
REVIEW RETURNED	28-Oct-2017

GENERAL COMMENTS	Overall, the article has a reasonable hypothesis and methodology. The authors conducted a meta-analysis for testing the association of ki67 expression with progression (PFS) and recurrence (RFS) in non-muscle invasive bladder cancer. Given conflicting literature on this association, the meta-analysis is important. The meta-analysis methods are well documented and followed the PRISMA guidelines. The study quality and heterogeneity were evaluated. The conclusion that ki67 is a predictor of PFS appears to be sound and the limitations are given. It is not clear whether PFS or RFS is more important, which impacts the clinical relevance of the findings. Issues:  1. The article has many errors with regards to the use of the English language, and requires extensive editing and correction. These errors sometimes obfuscated the meaning and occurred throughout the manuscript starting with the abstract. This made the article difficult to read and interpret. 2. The abstract does not state the hazard ratios or confidence intervals. 3. It is not clear whether ki67 is a dichotomous or a continuous
---

	predictor. If it is dichotomous, were all studies using the same threshold? 4. In the results on RFS, results for subgroups (Caucasian and shorter follow-up period) are reported, but it is not clear this subgroup analysis was part of the planned analysis or post hoc and driven by study heterogeneity. 5. The Galbraith plot is very consistent with the forest plot for identifying the outliers for RFS and PFS. The Galbraith plot axis labels could be more descriptive and rounded. 6. The number of studies considered in the final meta-analysis was 11, but only a subset of these were considered for PFS (9) and RFS (7). These sample sizes limit the potential analyses. 7. It would be valuable to add a discussion of the prognostic value in terms of the c-index or other measure of predictive accuracy. 8. Minor: Fonts on graphics could be larger.
--	---

REVIEWER	Dr Yana Vinogradova University of Nottingham United Kingdom
REVIEW RETURNED	28-Nov-2017

GENERAL COMMENTS	The paper is well-written and well-structured. I have only a few comments, which may help to improve the presentation and interpretation of the results. General comments: This is an observational study so it cannot answer the question whether ki67 actually influences the prognosis or if there is only an association. This should be made clearer – for example, see Page 9 line 45, or the subheadings in Results on pages 15 and 18. The authors have run 14 subgroup analyses (7 for RFS and 7 for PFS), but they still consider 5% to be statistically significant level. I think this should be reduced to 1% otherwise the probability of getting random spuriously-significant results is quite high. What were the rationales for these subgroup analyses? For example, what was the rationale for comparing shorter-term with longer-term studies? The same applies to the other subgroup analyses – what hypothesis was being tested in each case? The results for subgroup analyses in fact indicate that there were no differences between any subgroups, because all the confidence intervals overlap. In the abstract, the authors report as important the findings for Caucasian subgroup but in the light of there being no difference between this and the Asian subgroup, I think this is perhaps overstating the findings. Throughout the paper, expressions like “had a statistical significance” do not have much meaning. I would suggest the authors should more accurately use expressions like “had a statistically significant association” – also stating clearly whether these were with increased or decreased survival. Other comments are: Page 10 line 12: To make more sense, I suggest the wording “statistically significant difference”.
---

	Page 10 line 50: I would suggest using the word “considered” rather than “believed”. Page 18 line 8: This statement is wrong and needs correction.
--	--

VERSION 1 – AUTHOR RESPONSE

Responds to editorial requirements:

Comment 1: Please revise the Strengths and Limitations section (after the abstract) to focus on the methodological strengths and limitations of your study rather than summarizing the results.

Response: Thank you for the comments. We have rewritten the section of Strengths and limitations in the revised version.

Comment 2: Please work to improve the quality of English throughout the manuscript, either with the help of a native speaking colleague or with the assistance of a professional copyediting agency.

Response: The manuscript is revised according to the editors’ and reviewers’ comments, and utilized a professional copyediting agency to edit this revised version. The relevant certification can be verified at www.medsciediting.com/certificate/. The verification code is 0301-7583-6A68-2972-BEC6.

Responds to comments from the Associate Editor:

Comment 1: This paper would need to be sent to language polishing services as it was hard to read in places.

Response: Thank you for the comments from the Associate Editor. We improved the quality of English in the revised version with the help of a professional copyediting agency.

Comment 2: The conclusions section has duplication of sentences and where it reads “(...) and PFS had no statistical significance” I think they mean RFS: “For the patients with NMIBC treated with BCG intravesical immunotherapy, the overexpression of ki67 was the risk factor for PFS, the overexpression of ki67 was the risk factor for PFS, but the relationship between ki67 expression and PFS had no statistical significance. (...)”.

Response: The repetitive sentences and the incorrect expression in the “Conclusion” section are amended in the revised version. The correct conclusion is that the overexpression of Ki67 is the risk factor for PFS in patients with NMIBC after TUR and BCG intravesical immunotherapy; however, the relationship between the expression of Ki67 and RFS is not statistically significant. Owing to the limitations above of the present study, RCTs with large sample size are yet required to validate the results.

Comment 3: I agree with Gelfond that it would help the clinical community, specially those who are not experts in the area, to discuss what is the difference and the clinical relevance of PFS and RFS, and if there is one parameter more important than the other. What a shame we have good statistical reviews but no good clinical reviews. The authors acknowledge the study is not definitive so while I don’t think this will have major clinical implications for practice, it paves the way for future research needs. I also wondered whether they have a more detailed search strategy, since what they report seems thin.

Response: In the context of reviewing the clinical studies, we added a review section for clinical literature and discussed the potential clinical applications, including the of prediction of physician-to-patient prognosis and the corresponding implications for further clinical studies. Also, the search strategy was described in the revised version.

Responds to reviewer(s)' Comments:

Reviewer 1

Comment: Thank you for an interesting paper about prognostic value of ki67. There's nothing to comment about.

Response: We greatly appreciate Dr. Hanjong Ahn for reviewing this manuscript.

Reviewer 2

Comment: I have no comment for this paper.

Response: We would like to thank Dr. Ajili Faouzia for reviewing this manuscript.

Reviewer 3

General comment: This manuscript described results of a meta-analysis to explore whether ki67 is a prognostic factor of recurrence free survival (RFS) and progression free survival (PFS) in Bacillus Calmette–Guérin-treated non-muscle-invasive bladder cancer. Although the authors concluded that ki67 was not a prognostic factor of RFS but of PFS, several aspects of this manuscript should be improved to conclude that.

Response: We thank Kentaro Sakamaki. The corresponding amendments and explanations have been incorporated in the revised version.

Major comment 1: Hazard ratios (HRs) of univariate analysis or of multivariate analysis were used in the meta-analysis. The authors should clarify which HRs are unadjusted or adjusted in the manuscript, consider an influence of adjustments in meta-analysis and interpret results appropriately.

Response: Thank you for the comment. The univariate analysis showed that cancer-specific survival, disease-free survival, overall survival, PFS, and RFS were significantly correlated with poor prognosis in patients with a high reactivity of Ki-67. For RFS, all included data are multivariate, which account for the factors such as age and gender that affect heterogeneity, and the results of this adjustment should be accurate. For PFS, multivariate data were extracted from Oderda, Park, Quintero, Bertz, van Rhijn, and Santos: HR = 2.10, 95% CI = (1.07, 1.12), Z = 2.16, and P-value = 0.031. Data were extracted from Queipo-Zaragoza and Lopez-Beltran as Univariate, HR = 2.80, 95% CI = (1.65, 7.85), Z = 3.22, and P-value = 0.001. For combined effects, HR = 2.567, 95% CI = (1.562, 4.219), Z = 3.72, P-value < 0.001). The results showed a statistically significant difference in PFS. Overall, the above adjustments might not impact the meta-analysis. Consequently, in the revised version, the required modifications have been inserted in the related sections and shown in Table 4.

Major comment 2: It is unclear how factors in subgroup analyses and meta-regression were selected. For example, 'Cut off' was binarized as '15%' and 'Others', however 'Others' included such as '13%' and '40%'. The authors seemed not to consider 'Publication Time' in subgroup analyses but in meta-regression. And almost prognostic factors were not mentioned in the manuscript. Selection of factors depends on the HRs were adjusted or not. Please explain the validity of the subgroups analysis and the meta-regression.

Response: The responses to the comment are as follows:

- a. Factors in the subgroup analyses and meta-regression are selected based on the published articles and clinical experiences. Ki67 is dichotomous, and most of the included documents use an average threshold of 15%, which is selected as the cut-off value. Other subgroup factors include region, sample size, follow-up, stage, and patient age, which might be attributed to the following: 1. These factors are critical for the prognosis of the disease (except for the time of publication) in clinical practice; 2. From the design point of view, data with respect to the factors mentioned above were extracted from the included literature; 3. Reference also exhibits similar groupings. (p53 status correlates with the risk of recurrence in non-muscle invasive bladder cancers treated with BCG: a meta-analysis. PLoS One 2015; 10: e0119476. doi: 10.1371/journal.pone.0119476).
- b. Subgroup analysis of publication duration time has been added in the revised version.
- c. Prognostic factors are supplemented in the discussion section and discussed in conjunction with the article.
- d. The validity of the subgroups analysis is explained in the Major comment 4. Publication time is the reason for heterogeneity of PFS, which might be due to the advances in testing technology and increase in published articles worldwide.

Major comment 3: In order that HRs were valid, it is important for when and how ki67 was measured. The definitions of RFS and PFS are also important. If there was any difference, please clarify the measurements and the definitions in Methods and Results sections.

Response: The measurement of Ki-67 is according to the standard treatment option for NMIBC. The present study aimed to explore the prognostic value of Ki-67 as a marker in BCG-treated NMIBC. The meta-analysis was undertaken to evaluate the prognostic value of Ki-67 in patients with NMIBC after BCG therapy. Time and methods for the surgical removal of tumor tissue specimens can be followed pathologically. PFS is defined as the time from random assignment in a clinical trial to disease progression or death from any cause (RFS is defined as the time period from the date of randomization to the date of first documentation of recurrence (with cytological or histological confirmation or with radiological evidence) or death, whichever occurred first).

Major comment 4: Heterogeneity of HRs would be caused by many reasons, including Comment 1-3. The discussion on the heterogeneity would be poor. Please explain whether the heterogeneity could change the conclusion.

Response: The response to the comment is as follows:

a. The discussion on the heterogeneity has been supplemented in the revised version. We discussed the classification of heterogeneity based on clinical diversity, methodological diversity, and statistical diversity.

b. In order to ensure statistical homogeneity, we use I²-squared and P-values to evaluate the merging methods among studies. The relationship between Ki67 expression and RFS was not significantly different between studies (I² = 36.7%, P = 0.148). Also, statistical heterogeneity was between the studies (I² = 55.6%, P = 0.021) with respect to the relationship between Ki-67 expression and PFS. Although heterogeneity is observed in the studies, the combined data are clinically significant, and hence, we adopted the random-effects model to conduct the combined analysis.

c. In order to guarantee the methodological homogeneity, this study conducted a stringent quality evaluation on the methods that need to be merged, including randomized method, blinded implementation, whether there is a control, whether the control is reasonable, whether the randomized scheme is hidden, and whether there is a baseline similarity.

d. In order to ensure the clinical homogeneity and the consistency of the research object, and treatment factors, the study formulates rigorous and unified inclusion and exclusion criteria. Only those studies with the same research purpose and high-quality can be included in the analysis.

e. Based on the forest plot, although no overlap or heterogeneity of the confidence interval was observed between the studies, the point estimates were mostly on the right side of the ineffective line, and the effect of heterogeneity on the reliability of conclusions may not be significant.

In summary, the study strictly controlled the clinical and methodological heterogeneity, and no statistically significant heterogeneity was observed; thus, the heterogeneity of the included studies would not alter the results of this study.

Minor comments 1: Axes of Figure 4 were not correct. Please correct these axes.

Response: Thank you for pointing out the error. The vertical axis has been changed from se (hr) to se (ln hr).

Minor comments 2: Reference 27 is not appropriate. Please change the reference appropriately.

Response: We have replaced the reference 27.

Reviewer 4

General comment: Overall, the article has a reasonable hypothesis and methodology. The authors conducted a meta-analysis for testing the association of ki67 expression with progression (PFS) and recurrence (RFS) in non-muscle invasive bladder cancer. Given conflicting literature on this association, the meta-analysis is important. The meta-analysis methods are well documented and

followed the PRISMA guidelines. The study quality and heterogeneity were evaluated. The conclusion that ki67 is a predictor of PFS appears to be sound and the limitations are given. It is not clear whether PFS or RFS is more important, which impacts the clinical relevance of the findings.

Response: We thank Dr. Jonathan Gelfond, and have modified accordingly.

Comment 1: The article has many errors with regards to the use of the English language, and requires extensive editing and correction. These errors sometimes obfuscated the meaning and occurred throughout the manuscript starting with the abstract. This made the article difficult to read and interpret.

Response: A professional copyediting service was utilized to edit this manuscript.

Comment 2: The abstract does not state the hazard ratios or confidence intervals.

Response: in the correction has been made in the revised version.

Comment 3: It is not clear whether ki67 is a dichotomous or a continuous predictor. If it is dichotomous, were all studies using the same threshold?

Response: Ki67 is dichotomous, and the included studies use an average threshold of 15%, which is selected as the cut-off value. For both RFS and PFS, we performed a subgroup analysis simultaneously and found no difference between subgroups and overall results.

Comment 4: In the results on RFS, results for subgroups (Caucasian and shorter follow-up period) are reported, but it is not clear this subgroup analysis was part of the planned analysis or post hoc and driven by study heterogeneity.

Response: Before conducting this meta-analysis, the subgroup was set up during the protocol stage. The principles of the subgroup analysis were pre-set and reasonably set.

Comment 5: The Galbraith plot is very consistent with the forest plot for identifying the outliers for RFS and PFS. The Galbraith plot axis labels could be more descriptive and rounded.

Response: The Galbraith plot axis labels appropriately. The original "b" represented the HRs in the plots.

Comment 6: The number of studies considered in the final meta-analysis was 11, but only a subset of these were considered for PFS (9) and RFS (7). These sample sizes limit the potential analyses.

Response: Thanks to the reviewer for this suggestion. This is also a limitation of the current study. This meta-analysis was performed under stringent inclusion criteria, and those in the literature excluded a majority of the nonconforming literature. This has been illustrated in the "Strengths and limitations of this study" and "Conclusions". In the conclusion section of, we also explained that RCTs with large sample size are essential for the validation of the results.

Comment 7: It would be valuable to add a discussion of the prognostic value in terms of the c-index or other measure of predictive accuracy.

Response: Thank you for the suggestion. However, is it difficult to implement due to the following reasons: a. The statistical software used in this study is STATA, and c-index cannot be calculated via STATA.

b. The purpose of c-index is to assess the inconsistency; the I-squared and P-values are provided in the study that reflect the consistency of the data.

c. The present study employed statistical methods for calculating I-squared and P-values.

In summary, we examined the method of checking the consistency in this study. The calculation of c-index requires other software (for example R software). Changing the calculation software might give rise to other bias in the study results. Finally, if we have the opportunity to continue this research, we will discuss the difference in consistency predicted by R software and STATA.

Comment 8: Minor: Fonts on graphics could be larger.

Response: In the revised version, we replaced all the images according to the reviewer's comments.

Reviewer 5

General comment 1: This is an observational study so it cannot answer the question whether ki67 actually influences the prognosis or if there is only an association. This should be made clearer – for example, see Page 9 line 45, or the subheadings in Results on pages 15 and 18.

Response: Thank you Dr. Yana Vinogradova for reviewing the manuscript. The corresponding amendments have been made in the article.

General comment 2: The authors have run 14 subgroup analyses (7 for RFS and 7 for PFS), but they still consider 5% to be statistically significant level. I think this should be reduced to 1% otherwise the probability of getting random spuriously-significant results is quite high.

Response: 5% is used as a statistical significance level in the subgroup analysis as the possibility of obtaining randomly false results is high. The amendments have been made in the article.

General comment 3: What were the rationales for these subgroup analyses? For example, what was the rationale for comparing shorter-term with longer-term studies? The same applies to the other subgroup analyses – what hypothesis was being tested in each case?

Response: Thanks to the reviewer for pointing out the issue. The responses to the comments are as follows:

a. The choice of sub-group is mainly based on the following points: 1. Factors are critical for the prognosis of the disease (except for the time of publication) in clinical practice; 2. From the design point of view, these only data that can be extracted from the included literature; 3. The reference also has similar groupings. (p53 status correlates with the risk of recurrence in non-muscle invasive bladder cancers treated with BCG: a meta-analysis. PLoS One 2015; 10: e0119476. doi: 10.1371 / journal.pone.0119476).

b. The purpose of these subgroup analyses is to validate the interaction between subgroups of factors and elucidate the potential heterogeneity. The rationale for comparing the short-term and long-term studies is primarily to observe if subgroups of follow-up time influence the summary. The subgroup had a cut-off of 60 months, based on the average number of follow-ups included in the literature.

c. Other subgroups are chosen for a similar reason and purpose as the above subgroup.

General comment 4: The results for subgroup analyses in fact indicate that there were no differences between any subgroups, because all the confidence intervals overlap. In the abstract, the authors report as important the findings for Caucasian subgroup but in the light of there being no difference between this and the Asian subgroup, I think this is perhaps overstating the findings.

Response: Thank you the comment. The results of a single subgroup in a subgroup analysis is not a recommended conclusion. If subgroup analyses differ in subgroup outcomes, there is an interaction between subgroup factors, research hypotheses, and outcomes, suggesting that the significance of Ki67 for Caucasians and Asians may be different. However, Ki67 in Caucasians may not be deemed pivotal. These modifications have been made in the revised version.

General comment 5: Throughout the paper, expressions like “had a statistical significance” do not have much meaning. I would suggest the authors should more accurately use expressions like “had a statistically significant association” – also stating clearly whether these were with increased or decreased survival.

Response: It has been modified accordingly.

Other comments 1: Page 10 line 12: To make more sense, I suggest the wording “statistically significant difference”.

Response: It has been modified.

Other comments 2: Page 10 line 50: I would suggest using the word “considered” rather than “believed”.

Response: It has been corrected.

Other comments 3: Page 18 line 8: This statement is wrong and needs correction.

Response: It has been corrected.

VERSION 2 – REVIEW

REVIEWER	Yana Vinogradova University of Nottingham United Kingdom
REVIEW RETURNED	25-Jan-2018

GENERAL COMMENTS	I am generally happy with the revised manuscript. I have two comments regarding use of statistical terms. Pages 3, 13, 19: The authors inappropriately use the word "correlated/correlation". This is a statistical term allocated for relationship between two continuous variables. In this study, the authors refer to estimates derived from Cox model, therefore I suggest words "associated/association" instead. Page 15: Studies cannot be multivariate or univariate, analyses could be.
--

VERSION 2 – AUTHOR RESPONSE

Editorial Requirements:

Comment 1: Please continue to improve the quality of English throughout the manuscript, either with the help of a native speaking colleague or with the assistance of a professional copyediting agency.

Response: Thank you for the comments. The manuscript is revised according to the editors' and reviewer's comments, and reutilized a professional copyediting agency to edit this revised version.

Comment 2: Please include an example of a full electronic search strategy as a supplementary file, as per the requirements of the PRISMA checklist.

Response: We use the PubMed database as an example to summarize the search strategy as a supplementary file, according to the requirements of the PRISMA checklist

Reviewer: 5

Comment 1: Pages 3, 13, 19: The authors inappropriately use the word "correlated/correlation". This is a statistical term allocated for relationship between two continuous variables. In this study, the authors refer to estimates derived from Cox model, therefore I suggest words "associated/association" instead.

Response: Thank you Dr.Yana Vinogradova for reviewing the manuscript. It has been modified.

Comment 2: Page 15: Studies cannot be multivariate or univariate, analyses could be.

Response: It has been corrected.